# Acetyl-CoA Carboxylase (ACC) Inhibitor, CP640186, Effectively Inhibited Dengue Virus (DENV) Infection via Regulating ACC Phosphorylation

**DOI:** 10.3390/molecules27238583

**Published:** 2022-12-05

**Authors:** Wenyu Wu, Ruilin Chen, Yuanda Wan, Liren Li, Jiajia Han, Qiyun Lei, Zhipeng Chen, Shuwen Liu, Xingang Yao

**Affiliations:** 1State Key Laboratory of Ophthalmology, Zhongshan Ophthalmic Center, Sun Yat-sen University, Guangzhou 510060, China; 2State Key Laboratory of Organ Failure Research, NMPA Key Laboratory for Research and Evaluation of Drug Metabolism, Guangdong Provincial Key Laboratory of New Drug Screening, School of Pharmaceutical Sciences, Southern Medical University, Guangzhou 510515, China

**Keywords:** dengue virus, acetyl-CoA carboxylase, antiviral drugs, CP640186

## Abstract

Dengue fever is the most common mosquito-borne viral disease and is caused by the dengue virus (DENV). There is still a lack of efficient drugs against DENV infection, so it is urgent to develop new inhibitors for future clinical use. Our previous research indicated the role of VEGFR2/AMPK in regulating cellular metabolism during DENV infection, while acetyl-CoA carboxylase (ACC) is located downstream of AMPK and plays a crucial role in mediating cellular lipid synthesis; therefore, we speculated that an ACC inhibitor could serve as an antiviral agent against DENV. Luckily, we found that CP640186, a reported noncompetitive ACC inhibitor, significantly inhibited DENV proliferation, and CP640186 clearly reduced DENV2 proliferation at an early stage with an EC_50_ of 0.50 μM. A mechanism study indicated that CP640186 inhibited ACC activation and destroyed the cellular lipid environment for viral proliferation. In the DENV2 infection mice model, oral CP640186 administration (10 mg/kg/day) significantly improved the mice survival rate after DENV2 infection. In summary, our research suggests that lipid synthesis plays an important role during DENV2 proliferation and indicates that CP640186 is a promising drug candidate against DNEV2 in the future.

## 1. Introduction

Dengue fever is an acute infectious disease caused by the dengue virus and is transmitted mainly through mosquito vectors. After infection with the dengue virus, fever, muscle and joint pain, gastrointestinal bleeding, hemorrhagic shock, and even death can occur [1]. Today, the spread of dengue fever seriously threatens public health security, with around tens of millions of infections worldwide [2]. The dengue virus is mainly divided into four serotypes, and there are still no effective treatment drugs and vaccines [3]. Therefore, it is urgent to discover and develop new treatments or drugs against DNEV.

The dengue virus is a single-stranded positive-strand RNA (+ssRNA) virus, which initiates infection of a permissive cell via clathrin-mediated endocytosis and then releases its genomic RNA into the cytosol after fusing with the late endosome [4]. Various cells could be infected, such as the skin’s resident dendritic cells, monocytes and macrophages, megakaryocytes, erythroid precursor cells, liver cells, and endothelial cells. Several studies have revealed the relationship between lipid metabolism and viral replication during viral replication in cells [5,6,7]. We indicated the role of the VEGFR2/AMPK pathway in regulating cellular lipids [8], indicating the role of lipids in dengue viral infection. First, the enveloped virions need lipids as structural membrane components, with nearly 20% of the weight of the dengue virion being lipids [9]. Second, viral replication expression, replication, and even assembly always occur in cellular lipid bilayers to escape the host immune surveillance [10]. Third, β-oxidation is a normal way to provide energy for DENV replication in two ways, increasing autophagy and de novo synthesis [11]. Despite the numerous roles of autophagy in regulating cellular homeostasis, its regulation of lipid metabolism is a significant contributor to robust DENV replication [11,12]. DENV-induced autophagy stimulates the delivery of lipids to lysosomal compartments, releasing free fatty acids, which undergo β-oxidation in the mitochondria to generate ATP, producing a metabolically favorable environment for viral replication [11]. Lipid de novo synthesis is another way to provide energy, and it is reported that fatty acid synthase (FAS) re-localized to the replication complex by interaction with DENV NS3 protein, which increases the de novo synthesis of fatty acids during DENV infection [13]. Considering the above reasons, drugs targeting lipid metabolism are a promising way to inhibit DENV proliferation. Indeed, several small molecules regulating autophagy and lipid metabolism have been reported. Autophagy inhibitor 3-methyladenine (3MA) or siRNAs targeting autophagy gene expression compromised viral infection [11]. In addition, a small molecule inhibitor of FAS (C75) was reported to inhibit DENV replication in cell culture [14]. However, another FAS inhibitor (Orlistat) has weak antiviral activity and weak interaction between FAS and NS3 [15]. In addition, the small molecular compound PF-429242, known as an S1P inhibitor, is a promising candidate against DENV [5] and anti-Zika virus. Considering these reasons, it is reasonable to consider whether modulation of the upstream FAS pathway could inhibit dengue virus infection.

Fatty acid synthase action is started from palmitic acid, a 16-carbon fatty acid, and longer saturated fatty acids are synthesized by an elongation enzyme system based on palmitic acid [15,16]. Additional carbons are added in 2-carbon units (CO_2_ released) using malonyl-coenzyme A (malonyl CoA) as the carbon donor and different elongation enzymes. Malonyl CoA is generated from acetyl-CoA carboxylase (ACC), not only a substrate for de novo lipogenesis but also an inhibitor of mitochondrial fatty acid β-oxidation through inhibition of carnitine-palmitoyl transferase I (CPT-1), responsible for the transport of long-chain fatty acyl-CoAs across the mitochondrial membrane. ACC has two isoforms, ACC1 located primarily in the liver and adipose tissue and ACC2 dominant in skeletal and heart muscle. ACC inhibitors are hoped to inhibit de novo lipogenesis and increase the β-oxidation of long-chain fatty acids with the potential to treat of type 2 diabetes [16] and cancers [17]. The key regulatory role of ACC in fatty acid synthesis and oxidation pathways makes it an attractive target for various metabolic diseases [18]. In particular, the combination of ACC inhibitors with other drugs is a new strategy for the treatment of nonalcoholic fatty liver disease and nonalcoholic steatohepatitis [19]. Expanding the clinical indications for ACC inhibitors will be one of the main research directions in the future.

CP640186 is a reported potent inhibitor of mammalian ACCs (Figure 1A), which was discovered by researchers at Pfizer Inc. with IC_50_ values of about 55 nM [20]. CP640186 could reduce tissue malonyl-CoA levels, inhibit fatty acid biosynthesis, and stimulate fatty acid oxidation. In addition, CP640186 could reduce body fat mass and body weight and improve insulin sensitivity. It inhibited both isozymes with an IC_50_ of 55 nM in inhibiting HepG2 cell fatty acid and TG synthesis. CP640186 also stimulated fatty acid oxidation in C2C12 cells (ACC2) and in rat epitrochlearis muscle strips with an EC_50_ of 57 nM and 1.3 μM. Kinetic studies showed that CP640186 is noncompetitive versus the acetyl-CoA substrate but may function at the active site of the CT domain [20]. At present, a variety of ACC inhibitors based on the structure of CP640186 have been developed for the treatment of metabolic diseases, so we mainly use CP640186 in our research. Recently, two groups have reported the role of ACC in West Nile virus (WMV) replication, and ACC inhibitors (PF-05175157, PF-05206574, PF-06256254, and 5-(tetradecyloxy)-2-furoic acid (TOFA)) were shown to have an impressive effect against WMV in vitro and in vivo [21,22]. Teresa et al. reported the effect of the ACC inhibitor 5-(tetradecyloxy)-2-furoic acid (TOFA) on infection by WNV. Treatment with TOFA significantly reduced the cellular content of multiple lipids and inhibited the multiplication of WNV in a dose-dependent manner. They also found that another ACC inhibitor, 3,3,14,14-tetramethylhexadecanedioic acid (MEDICA 16), also inhibited WNV infection, pointing to the ACC as a druggable cellular target suitable for antiviral development against West Nile virus (WNV) [22]. Nereida et al. reported the effect of three small-molecule ACC inhibitors (PF-05175157, PF-05206574, and PF-06256254) on the infection of WNV, dengue virus, and Zika virus. They found that PF-05175157 induced a reduction in the viral load in serum and kidney in WNV-infected mice, unveiling its therapeutic potential for treating chronic kidney disease associated with persistent WNV infection [21]. Another group reported that the cholesterol-enriched cellular environment is crucial for viral replication, and they found that metformin and lovastatin (HMGCR inhibitor) inhibited the dengue virus proliferation [23]. These results support the repositioning of metabolic inhibitors as broad-spectrum antivirals [21,23,24]. However, the detailed mechanism of ACC against DENV need further study.

The above study indicated that ACC might participate in the DENV proliferation process. However, whether CP640186 could suppress DENV proliferation in vivo still needs to be studied. In this study, we applied CP640186 and DENV New Guinea C strain (NGC) as tools to disclose the antiviral ability between ACC and dengue viral proliferation, and also to evaluate the antiviral ability of CP640186 against DENV in vivo.

## 2. Results

### 2.1. The Antiviral Effect of CP640186

To study the anti-DENV effect of ACC inhibitors, CP640186 (Figure 1A) was incubated with BHK-21 cells after 1 h DENV2 infection, and then the antiviral effects were observed under a microscope through the classic cytopathic assay. CP640186 showed a clear antiviral ability at 1 μM (Figure 1B). From the data of the cytopathic assay, the cellular cytopathic effect was observed clearly after DENV2 infection (Figure 1B), thus reducing cellular viability. With CP640186 incubation, the amelioration of cytopathic effects could be observed at a low dose of 0.25 µM, and the cytopathic effect was hard to be observed at a high concentration of 2 µM (Figure 1B). These data indicated the clear antiviral ability of CP640186. We next evaluated the antiviral ability of CP640186 against four serotypes of DENV through a cellular viability assay. As shown in Figure 1C, CP640186 has a similar antiviral effect and activity against four serotypes of DENV, with IC_50_ of 0.96 µM, 1.22 µM, 0.99 µM, and 1.69µM separately on BHK-21 cells. We also verified its effect on another flaviviridae virus, the Zika virus; data indicated a strong antiviral ability of CP640186 against the Zika virus through cytopathic assay (Figure 1D), with an IC_50_ of 1.27 µM (Figure 1E). We further tested the cellular toxicity of CP640186 on BHK-21 cells by GTP assay, indicating that CP640186 did not affect the cell viability even at a dose higher than 10 times of the IC_50_ dose (20 µM) (Figure 1F). These results showed that CP640186 has a good antiviral ability and low toxicity.

### 2.2. The Antiviral Ability of CP640186 Was via ACC Phosphorylation

In order to clarify the role of CP640186 in regulating lipid metabolism, we detected the phosphorylation level of ACC after DENV2 infection in the presence of CP640186 by Western blotting. The results showed that the phosphorylation level of ACC was down-regulated after virus infection, while its phosphorylation level was up-regulated in a dose-dependent manner by CP640186 treatment (Figure 2A and Appendix A). It is noteworthy that with 1 µM CP640186 incubation, the phosphorylation of ACC was significantly up-regulated. At the same time, viral structural protein E was also attenuated, along with the increased phosphorylation of ACC. In order to analyze the effect of CP640186 on cellular lipid metabolism, an oil red staining assay was used to detect the cellular lipid droplets [8]. No lipid droplets were observed in cells after DENV2 infection, suggesting that the virus infection depleted cellular lipid droplets (Figure 2B). In contrast, the cellular lipid droplets increased with the incubation of CP640186 in a dose-dependent manner (Figure 2B). Although viral E protein was hardly observed with 2 μM CP640186 treatment (Figure 2A), the cellular lipid droplet level did not return to the normal level under this condition (Figure 2B). These results showed that dengue virus replication required the participation of cellular lipids, and it is feasible to inhibit viral infection via regulating lipid droplet formation through CP640186.

Then, we evaluated the effect of CP640186 on the viral protein transcriptional and translational levels. After BHK-21 cells were infected by DENV2 for 1 h, the virus was washed and replaced with a medium containing CP640186 at indicated concentrations. CP640186 treatment significantly reduced the mRNA level of viral E and NS1 protein compared to the control group (Figure 2C). Under confocal microscopy, it was observed that the level of viral E protein was significantly reduced after CP640186 treatment (Figure 2D,F). At the same time, we observed that the dsRNA level of the virus was reduced largely, indicating that the viral replication process was also inhibited by CP640186 (Figure 2E,F). Furthermore, we investigated the number of progeny viruses by plaque assay and found that CP640186 significantly reduced the number of progeny viruses at 1 μM (Figure 2G). It was shown that CP640186 inhibited the virus infection in the whole infection process.

### 2.3. The Dosage of Virus Affecting the Antiviral Effect of CP640186

Considering the antiviral effect of CP640186, we wanted to explore the antiviral effect under different virus dosages. Then, the antiviral effect of CP640186 was evaluated on the cells with different doses of virus infection. Three doses of viruses were applied with ten time intervals: 10^0^TCID_50_, 10^1^TCID_50_, and 10^2^TCID_50_. The results showed that as the inoculum of the virus (number) increased, the level of ACC phosphorylation gradually decreased, and it was accompanied by the increase in virus E protein (Figure 3A and Appendix A), while the effect of virus on ACC phosphorylation was reversed in the group with CP640186 incubation, accompanied with the reduction in E protein expression (Figure 3A). Indeed, compared with the low viral dosage group (10^0^TCID_50_ or 10^1^TCID_50_), we observed that the high virus dosage (10^2^TCID_50_) infection significantly weakened the effect of CP640186 on ACC phosphorylation and E protein expression, and the above phenomena were also confirmed by RT-qPCR (Figure 3B). We also observed similar phenomenon through the number of progeny virus production (Figure 3C). Compared with the low virus dosage infection group, the cells produced a higher sub-virus after high virus dosage infection with the CP640186 treatment. The above results suggested that the antiviral effect of CP640186 is indeed related to the dosage of the virus at the time of infection. From the above results, CP640186 could significantly reduce dengue virus infection accompanied by the increase in ACC phosphorylation.

### 2.4. The Antiviral Model of CP640186

Given that CP640186 owned a solid antiviral ability, we wanted to study the antiviral model of CP640186. A drug-addition assay was applied as we previously reported (Figure 4A) [25], and the antiviral effect of CP640186 was detected by viral protein, mRNA, and progeny virus titration. We set up four models to verify the antiviral mechanism of CP640186. First, CP640186 was incubated with the cell for 1 h before 1 h infection of cells with the DENV2 virus, which was used to determine whether the CP640186 would inhibit the virus entry. The viral E protein level by Western blotting indicated that CP640186 could not inhibit the viral protein expression (Figure 4B and Appendix A), indicating that CP640186 could not inhibit the virus entry. Next, CP640186 was co-incubated with the DENV2 virus during 1 h of infected cells to determine whether CP640186 could inactivate the virus (Figure 4B and Appendix A). There was no difference between the CP640186 treatment and no treatment group via viral E protein expression, suggesting that CP640186 did not affect the virus. Thirdly, cells were treated with CP640186 post DENV2 infection, which was used to observe whether the CP640186 acted on the host targets to play an antiviral effect after infection. The viral E protein was clearly reduced in this drug-addition model (Figure 4B and Appendix A). This result showed that CP640186 played antiviral effects post virus infection, and co- and post-drug-addition assay further confirmed this phenomenon. In order to further verify the above results, RT-qPCR was used to quantify the transcriptional level of viral NS1 and E protein. The result showed that CP640186 addition after virus infection could significantly inhibit viral gene transcription (Figure 4C). In addition, similar results were observed by the plaque formation assay (Figure 4D), showing that the number of plaque (blue dots) was reduced in the post-drug-addition model (Figure 4D), suggesting CP640186 exerted the antiviral effect after virus entry.

Then, we tested whether dosing at different times could affect the antiviral effect of CP640186. CP640186 was added into the supernatant of cells at different time points after DENV2 infection, and then the antiviral effect of CP640186 was verified by Western blotting assay (Figure 4E). Results showed that CP640186 addition within 8 h after virus infection induced ACC phosphorylation and reduced the level of virus E protein expression. However, CP640186 addition after 12 h could not induce the ACC phosphorylation nor inhibit the expression of virus E protein. These results were further confirmed by the gene transcriptional results by RT-qPCR level (Figure 4F and Appendix A). Therefore, these results indicated that CP640186 exerted its antiviral effects in the early stages of the virus entry (8 hpi).

### 2.5. The Antiviral Effect of CP640186 In Vivo

In order to study the antiviral effect of CP640186 in vivo, we used the previously established mice infection model with DENV2 [8]. IFNAR^−/−^C57Bl/6 (B6) mice were infected with DENV2 (10^6^ plaque-forming unit) by intraperitoneal injection. These mice have a lack of type-I interferon receptor and are widely used in studies of virus pathogenesis for the DENV2 infection (Figure 5A). After DENV2 infection, the mice could freely obtain food and drinking water. In the oral saline treatment group, all mice died between the 7th and 8th day compared with the uninfected group. However, in the oral CP640186-treated group, the mice survived to varying degrees. In the 2 mg/kg/day oral treatment group, the survival rate of mice was 34%. In the 5 mg/kg/day oral treatment group, the survival rate of mice was 66%. In the 10 mg/kg/day treatment group, nearly 84% of mice survived (Figure 5B). These data reflected the robust antiviral effect of CP640186 in vivo. According to the body weight of mice, we found that the weight loss of the mice after infection decreased rapidly. Compared with normal mice without DENV2 infection, we detected a slight change in mice body weight with CP640186 (10 mg/kg/day) administration (Figure 5C). Immediately after the mouse was sacrificed, fresh tissue was removed for the following analyses. We tested the virus number in the serum through the FFA assay (Figure 5D). Results showed that the quantity of virus number was dose-dependently dropped with the CP640186 treatment group. ACC phosphorylation level and viral E protein level were further studied in liver tissue through Western blotting, and the ACC phosphorylation level was up-regulated with the increase in CP640186 concentration (Figure 5E and Appendix A). We also observed similar results in other organs, as shown in Appendix A. At the same time, viral E protein was decreased with CP640186 administration (Figure 5E). We also detected virus infections in a variety of organs through RT-qPCR. After the drug administration, the virus genes in the organs decreased significantly (Figure 5F). These results indicated that CP640186 does work by affecting the ACC pathway. In summary, CP640186 has significant anti-DENV2 effects in vivo.

We also detected changes in inflammatory factors after virus infection, because cytokine storms are important causes of death with virus infection. The ELISA kit was used to measure the concentrations of inflammation-related factors (IFN-γ and IFN-β) in the mouse serum. As a result, the inflammatory factors in the drug-treated mice were clearly reduced (Figure 6A). The RT-qPCR assay showed that the decreased transcription levels of various inflammatory factors (IFN-1β, IL-1β, and CXCL10) in the liver, spleen, lung, and kidney tissue of the mice significantly reduced with increasing drug concentration treatment in the drug group (Figure 6B–D). In addition, we analyzed the character of inflammatory cell infiltrates by HE staining (blue nuclei of cells) in liver tissue (Figure 6E). The liver tissues of mice infected with DENV2 showed severe inflammatory infiltration. After CP640186 treatment, inflammatory infiltration was significantly reduced.

## 3. Discussion

Cellular lipid metabolism is closely related to dengue virus infection, as shown our and other groups’ reports [5,6,8,22,24,26]. Since ACC plays a rate-limiting role in lipid synthesis, we think ACC inhibitors could be promising antivirals against DENV. However, there are currently no in vivo data reported about the ACC inhibitors against dengue virus. The main reason is that the antiviral effect of the reported ACC inhibitor is not good enough, and there is also a lack of in vivo models. To address this issue, we found CP640186, with good antiviral activity and low toxicity. DENV2 proliferation could reduce by about 84.42% with CP640186 treatment at 1 μM with an EC_50_ of 0.5 µM, which is basically the lowest effective concentration in cell experiments. At the same time, we validated the antiviral effect of CP640186 in knock-out mice (B6 background mice). It is worth noting that CP640186 showed good antiviral activity in vivo, which could resist death with viral infection at a dosage of 10 mg/kg/day orally administrated. Our study comprehensively demonstrated the antiviral effect of CP640186 against the dengue virus in vivo and in vitro, showing that CP640186 is a promising compound and could be used in clinical applications. In a word, this is the first report of an ACC inhibitor with in vivo anti-DENV activity.

Although several studies disclosed the antiviral effect of an ACC inhibitor against flaviviruses, particularly West Nile virus, its underlying mechanism is still unclear. However, our results could explain the above phenomenon to a certain extent. Under normal circumstances, ACC inhibition will result in a decrease in intracellular lipids. However, we found that CP640186 retained cellular lipids during dengue virus infection. In other words, dengue virus infection consumed cell lipid droplets, and the CP640186 inhibited this process. This phenomenon raised two possibilities: firstly, that CP640186 inhibited dengue virus replication, which in turn affected dengue virus consumption of lipids; and secondly, that ACC inhibition leads to the loss of a key lipid in the process of dengue replication, which in turn inhibits the consumption of lipids by dengue. Based on our results, a large amount of cellular lipids was consumed, indicated by the oil red stain, especially when the inoculum of the virus (number) was increased. These results support the latter hypothesis, that CP640186 treatment leads to the loss of a key lipid in the process of dengue replication. As for the detailed mechanism of CP640186, we currently believe that it is mainly due to the inactivation of AMPK during dengue virus infection [8], which thus decreased the phosphorylation of ACC and increased the activity of ACC. Meanwhile, inhibition of ACC could increase the phosphorylation of ACC and reduce the activity of ACC. We added these descriptions to the discussion section.

Lipids are widely used during dengue virus proliferation and viral assembly [14]. Therefore, we attempted to clarify which process was affected by CP640186. Previous results indicated that lipid inhibition impaired membrane rearrangements associated with viral replication. In addition, lipid rafts combined with a cellular receptor are related to the entry of the dengue virus into cells [10]. However, we found that giving CP640186 treatment before virus infection did not inhibit viral replication, so this hypothesis was ruled out. Next, there are reports indicating that cellular lipids are related to viral immune escape. In this study, we found that CP640186 cannot change the innate immune pathway no matter when the virus infected or with CP640186 acting alone. These results indicated that ACC did not affect the natural immune response pathway, and it could not enhance the innate immune response, nor immune escape. In addition, lipid inhibition could alter the viral envelope composition and impair viral particle assembly. In fact, we found that CP640186 played a role in the early stages of viral replication. In addition, dysfunction of lipid synthesis could cause the generation of an unfavorable metabolic environment for virus replication. We did observe that CP640186 affected virus replication in the early stages of viral infection (1–4 hpi), suggesting that CP640186 might create an unfavorable environment for virus replication, which thus affected the viral assembly.

Given that inhibition of ACC activity could alter lipid metabolism, CP640186 was commonly used in studies of metabolic diseases such as hypertension, diabetes, visceral obesity, hyperlipidemia, cancer, fungal and bacterial infections, and herbicide exposure [20]. ACC is an enzyme that controls the rate-limiting reaction of long-chain fatty acid synthesis and mitochondrial fatty acid oxidation [26,27]. CP640186 inhibits the synthesis of fatty acids and the oxidation of fatty acids, so that various lipids are lacking during the replication of DENV, which in turn affects the generation of progeny viruses. Due to the rapid onset of dengue fever and the short duration of medication, the side effects of the drug can be ignored, because these side effects are often caused by long-term medication.

The doses of CP640186 used in vivo were different, which inhibited fatty acid synthesis in rats, CD1 mice, and ob/ob mice with ED_50_ of 13, 11, and 4 mg/kg, and stimulated rat whole-body fatty acid oxidation with an ED_50_ of 30 mg/kg. Taken together, the concentration used in this study was consistent with the previous report. Recently, several groups have disclosed the relationship between lipid metabolism and flavivirus infection [21,23,24]. Our research provided a new ACC inhibitor, CP640186, and studied the details of the antiviral effect in vitro and in vivo.

## 4. Materials and Methods

### 4.1. Materials

Baby Syrian hamster kidney cell line (BHK-21) was cultured in RPMI-1640 (Thermo Fisher Scientific, Waltham, MA, USA) supplemented with 10% fetal calf serum (Thermo Fisher Scientific, Waltham, MA, USA) in a 37 °C incubator with 5% CO_2_. C6/36 mosquito larva cells and DENV2 (New Guinea C derivative strain) were provided by Professor Xiaoguang Chen. DENV2 was amplified in C6/36 cells and stored at −80 °C until use. Professor Wei Zhao from Southern Medical University provided DENV1, 3, and 4. CP640186 were purchased from SuperLan chemical (Shanghai, China). MTT was purchased from Beyotime Biotechnology (Shanghai, China) (CAS: 97062-376, purity 98%).

### 4.2. The Cytopathic Effect (CPE) Assay

BHK-21 cells were incubated in a 96-well plate (5 × 10^4^ cells/well) overnight, washed twice with PBS, and DENV (10^1^TCID_50_) was seeded with 100 μL/well and incubated at 37 °C for 1 h. Then, the virus was removed, and the cells were washed twice with PBS. RPMI1640 containing CP640186 hydrochloride (0–2 µM) was added and the culture was continued for 4 days. The CPE on the cells was observed under a microscope (Nikon, Tokyo, Japan) [23]. The cellular viability was analyzed by MTT (3-[4,5-dimethylthiazol-2-yl]-2,5-diphenyl) assay.

### 4.3. Quantitative Real-Time PCR (RT-qPCR)

BHK-21 cells were first cultured in a 12-well plate (10^5^ cells/well). The cells were then infected with DENV2 (10^1^TCID_50_) for 1 h. Cellular RNA was extracted and purified using a viral RNA extraction kit (Qiagen, Dusseldorf, Germany). Then, cDNA was synthesized and amplified with a SuperScript-III kit (Takara, DaLian, China). The primer sequences are as follows. GAPDH primers were used as a reference.

DENV2 E protein Forward: 5′-GAGGGGAGCGAAGAGAATGG-3′

DENV2 E protein Reverse: 5′-GCCCCATAGATTGCTCCGAA-3′

GAPDH Forward: 5′-TGACCTCAACTACATGGTCTACA-3′

GAPDH Reverse: 5′-CTTCCCATTCTCGGCCTTG-3′

ISG56 Forward: 5′-ACACCTGAAAGGCCAGAATGAGGA-3′

ISG56 Reverse: 5′-TGCCAGTCTGCCCATGTGGTAATA-3′

DENV2 NS1 Forward: 5′-GGCATTTGTGGAATCCGCTC-3′

DENV2 NS1 Reverse: 5′-AGAGCATTTTCGCTTTGCCC-3′

IFN-β Forward: 5′-TTGTTGAGAACCTCCTGGCT-3′

IFN-β Reverse: 5′-CAGGTAATGCAGAATCCTCCCA-3′

IL-1β Forward: 5′-GTGTGGATCCCAAACAATACCC-3′

IL-1β Reverse: 5′-AAGACAGGTCTGTGCTCTGC-3′

IFITM1 Forward: 5′-GTTACTGGTATTCGGCTCTG-3′

IFITM1 Reverse: 5′-GGTGTGTGGGTATAAACTGC-3′

ISG15 Forward: 5′-GGTGTCCGTGACTAACTCCAT-3′

ISG15 Reverse: 5′-TGGAAAGGGTAAGACCGTCCT-3′

ISG54 Forward: 5′-GACACGGTTAAAGTGTGGAG-3′

ISG54 Reverse: 5′-GGTACTGGTTGTCAGGATTC-3′

CXCL10 Forward: 5′-CCAAGTGCTCCGTTTTTC-3′

CXCL10 Reverse: 5′-GGCTCGCAGGGATTTCAA-3′

IFITM3 Forward: 5′-ATGTCTCTGCCGTC-3′

IFITM3 Reverse: 5′-GTCATGAGGATGCCCAGAAT-3′

### 4.4. Western Blotting

E antibody was purchased from Neo bioscience Company (GTX127277, Beijing, China), and J2 (English & Scientific Consulting Kft, Budapest, Hungary), IFR3(11904S), p-IRF3 (29047S), β-actin (4970S), p-STAT1 (9167S), STAT1 (14994S), p-STAT2 (88410S), and STAT2 (72604S) were from Cell Signaling Technology (Boston, MA, USA). Briefly, cell lysate was separated by SDS-PAGE and transferred into the PVDF membrane (Amersham Biosciences, Buckinghamshire, UK). After incubation with corresponding primary antibodies overnight at 4 °C, the membrane was incubated with PBST with 3% BSA diluted secondary antibody (goat anti-mouse or goat-anti-rabbit IgG-HRP (1:5000)) for 0.5 h at room temperature. Finally, the membranes were visualized using the Dura detection system (Thermo Fisher Scientific, Waltham, MA, USA).

### 4.5. Determination of Viral Dosage (Plaque Focus Forming Assay, FFA)

BHK-21 (1 × 10^5^ cells/well) was incubated in a 96-well plate overnight. The virus was diluted, and the cells were infected for 1 h. The cells were fixed with 1.2% CMC and incubated for 3–4 days. After observing the lesion under a microscope, cells were fixed by adding 1% polyoxymethylene. The cells were perforated with PBS with 0.1% Triton X–100 (PBST). It was then blocked with PBST with 3% BSA for 1 h. The primary antibody (Millipore, MAB10216 (1:5000)) was incubated for 1 h at 37 °C, followed by incubation with PBST 3% BSA diluted secondary antibody (goat anti-mouse IgG-HRP (1:5000)) for 1 h at 37 °C. After washing with PBST 3 times, the cells were incubated with TMB on a shaker for 20 min, and the plate was washed and observed under a microscope to calculate the number of spots [8].

### 4.6. Confocal Microscopy

BHK-21 cells were plated at 5 × 10^4^ cell/mL on confocal dishes and were treated with different concentrations of CP640186 for 48 h. Then, the cells were fixed with 4% (*w/v* paraformaldehyde and were treated with 0.1% PBS (*v/v*) of Triton X-100 and were flow stained with primary antibody (2 μg/mL) at 4 °C overnight. Then, these cells were washed 3 times with PBS, fluorescein isothiocyanate or tetramethylrhodamine isothiocyanate-conjugated secondary antibody (Sigma, St. Louis, MO, USA) was added and incubated at 25 °C for 2 h. Then the cell nucleus were stained with HOCHEST33285 for 20 min. After washing 3 times with PBS, the image was obtained by a confocal microscope (Zeiss, Jena, Germany).

### 4.7. Oil Red Stain

An amount of 0.5 g of ORO (s-O0625, purchased from Sigma–Aldrich Company, Darmstadt, Germany) was added to 120 mL of 99% (*v*/*v*) isopropanol, and then 80 mL of H_2_O was added. The solution was then filtered through a 0.45 μM filter to remove the precipitates after stirring. The cells were washed twice with PBS and fixed with 4% paraformaldehyde (DF0135, purchased from LEAGEN, Shanghai, China) for 20 min. Then, the paraformaldehyde was washed away with PBS. The ethanol was then sucked out, and oil red dye solution was directly added, enough to cover the orifice, at room temperature for 30 min. The cells were washed with PBS 3 times after the red dye solution evaporated. Then, the cells were incubated with Hoechst (40729ES10) from Yeasen Biotechnology Co., Ltd., (Shanghai, China) for 20 min, and the nuclei were stained (PBS containing 3% BSA). The staining was observed under the confocal microscope as described in the previous report [8] (Zeiss, Jena, Germany).

### 4.8. Drug- and Time-Addition Assay

BHK-21 cells were incubated in 12-well plates (10^5^ cells/mL) overnight. As a drug-addition assay, the cells were infected with DENV2 (10^1^TCID_50_) for 1 h. As in our previous report [27,28,29,30], dengue virus was incubated with CP640186 in different models, and the viral protein was analyzed. As a time-addition assay, 10 µM of CP640186 was added to the infected cells at 0, 4, 8, and 12 h post infection (hpi). After 24 hpi, cellular RNA or protein was analyzed.

### 4.9. ELISA

The levels of IFN-β and IFN-γ in mouse serum were evaluated by IFN-β (EMC016, NEOBIOSCIENCE, Beijing, China) and IFN-γ (EK0375, BOSTER, Pleasanton, CA, USA) ELISA kit. Standards and samples were measured in triplicate. The optical density of the final color-developed sample was measured at 450 nm using the Infinite M1000 Pro (Tecan, Männedorf, Switzerland). The parameters of the “one site–total binding” equation were fitted to the 8-point standard curve to obtain the cytokine concentration using GraphPad Prism 8.0 software (GraphPad Software, San Diego, CA, USA).

### 4.10. Animal Experiments

This study was approved by the Animal Care and Use Committee of Southern Medical University (2019054) and conducted according to the care, use, and protocol of experimental animals. CP640186 was diluted in PBS at the appropriate concentration. To investigate the effects of CP640186 on the DENV2 infection, IFNAR^−^/^−^C57Bl/6 (B6) mice, gifts from Prof Jincun Zhao, were infected with DENV2 (10^6^ plaque-forming units) by intraperitoneal injection. The mice were divided into four groups of six. Mice in each group were orally administered 2, 5, or 10 mg/kg CP640186 or PBS once daily for six days. The mice were sacrificed at the end of the experiment and tissues were obtained for further analysis.

### 4.11. Statistical Analysis

Differences between treatments and control groups were evaluated using the Prism 5 software (GraphPad Software, San Diego, CA, USA) In all cases, parametric or nonparametric tests and the appropriate post hoc test were applied. If data conformed to the normality and equivariance (parametric) of the analysis of variance (ANOVA) hypothesis, a one-way ANOVA was performed followed by a Holm–Sidak multiple comparison post hoc test. Instead, multiple comparisons were performed by a Kruskal–Wallis one-way ANOVA on ranks followed by Dunnett multiple comparisons post hoc test for the data that did not meet ANOVA assumptions (nonparametric). In addition, the Student’s t-test was conducted for some cases. All the data are expressed as the mean ± standard deviation (SD), and *p* < 0.05 was regarded statistically significant. Values are presented as follows: ns *p* > 0.05, * *p* < 0.05, ** *p* < 0.01, and *** *p* < 0.001.

## Figures and Tables

**Figure 1 molecules-27-08583-f001:**
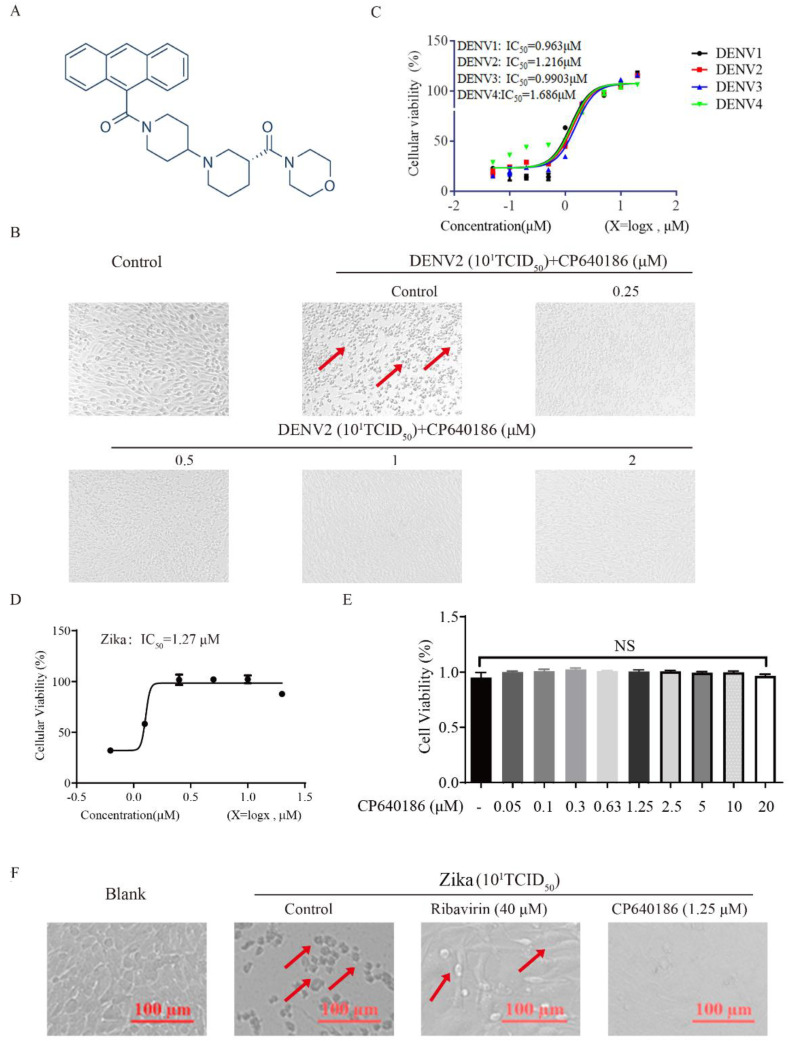
The antiviral ability of CP640186 against DENV2 and Zika. (**A**) The chemical structure of compound CP640186. (**B**) Cytopathic effects were observed after DENV2 infection in the presence of CP640186 at the indicated concentrations. (**C**) The IC_50_ of CP640186 against DENV1–4. (**D**) The IC_50_ of CP640186 against Zika virus. (**E**) The cytotoxicity effect of CP640186 on BHK-21 cells. (**F**) The cytopathic effect of CP640186 against Zika in Vero cells. Each experiment was repeated 3 times and column data represent mean ± SD.

**Figure 2 molecules-27-08583-f002:**
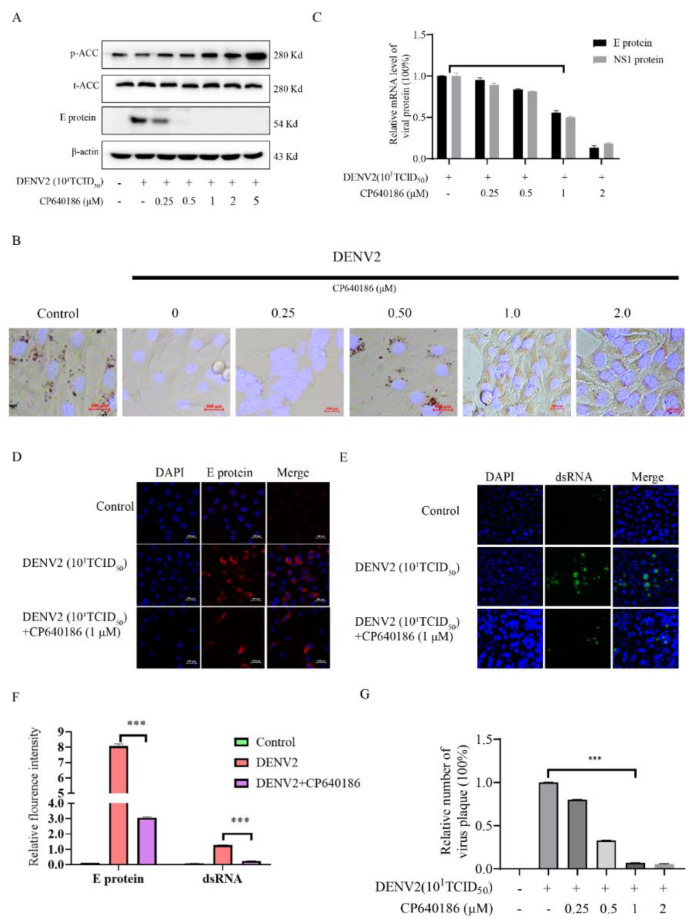
The relationship between ACC phosphorylation and DENV2 infection. After DENV2 infection of BHK-21 cells for 1 h, CP640186 was added at indicated concentrations and incubated for 48 h. (**A**) The levels of p-ACC, T-ACC, viral E protein, and GAPDH were analyzed by Western blotting. (**B**) The cellular lipid droplet was observed under microscope after oil red staining. (**C**) The mRNA levels of viral E protein and NS3 protein were analyzed by RT-qPCR assay. (**D**) The levels of viral E protein and (**E**) the levels of viral dsRNA were observed under confocal microscopy. (**F**) The quantification results of (**E**,**F**). (**G**) The number of progeny viruses in the supernatant was qualified by plaque assay. Each experiment was repeated 3 times and column data represent mean ± SD. *** indicates *p* < 0.001.

**Figure 3 molecules-27-08583-f003:**
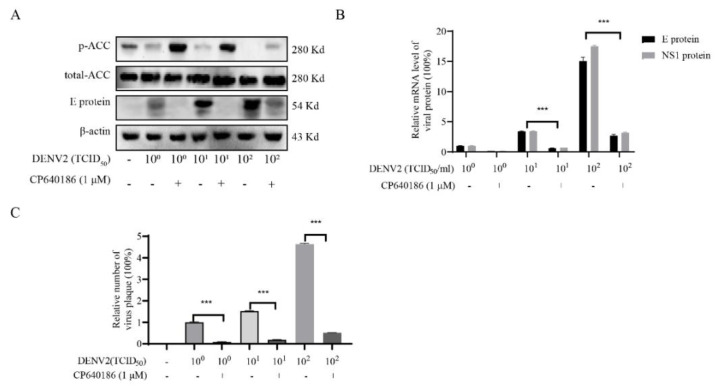
The effect of virus dosage on the antiviral effect of CP640186. After 1 h infection with different dosages of DENV2 (10^0^TCID_50_, 10^1^TCID_50_, and 10^2^TCID_50_), BHK-21 cells were incubated with CP640186 at 1 μM for 48 h. (**A**) Then, the protein levels of p-ACC, T-ACC, and E protein were analyzed by Western blotting. (**B**) The mRNA levels of viral E protein and NS3 protein were analyzed by RT-qPCR assay. (**C**) The number of progeny viruses in the supernatant was qualified by plaque assay. Each experiment was repeated 3 times and column data represent mean ± SD. *** indicates *p* < 0.001.

**Figure 4 molecules-27-08583-f004:**
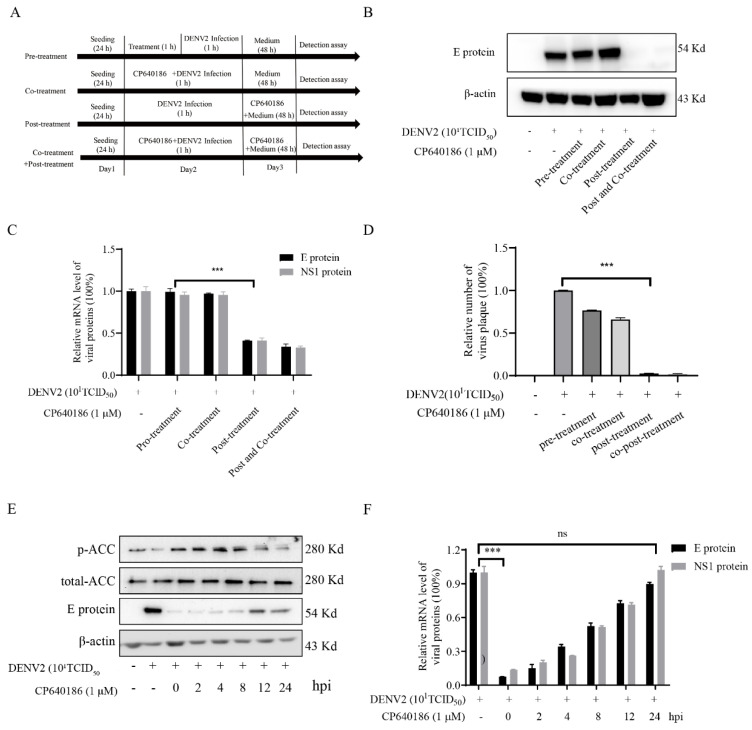
The antiviral model of CP640186. (**A**) The schematic diagram of drug addition of CP640186 with DENV2. (**B**) The viral E protein was analyzed by Western blotting, as shown in (**A**). (**C**) The mRNA levels of viral E protein and NS3 protein were analyzed by RT-qPCR assay, as shown in (**A**). (**D**) The number of progeny viruses in the supernatant was qualified by plaque assay, as shown in (**A**). (**E**) BHK-21 cells were infected with DENV2 at indicated times: 2 h, 4 h, 6 h, 8 h, 12 h, and 24 h. Immediately, the cell culture medium was replaced with a medium containing 1 μM CP640186. Then, the protein levels of p-ACC, T-ACC, and viral E protein were analyzed by Western blotting. (**F**) The mRNA levels of viral E protein and NS3 protein were analyzed by RT-qPCR assay, as shown in (**E**). Each experiment was repeated 3 times and column data represent mean ± SD. *** indicates *p* < 0.001.

**Figure 5 molecules-27-08583-f005:**
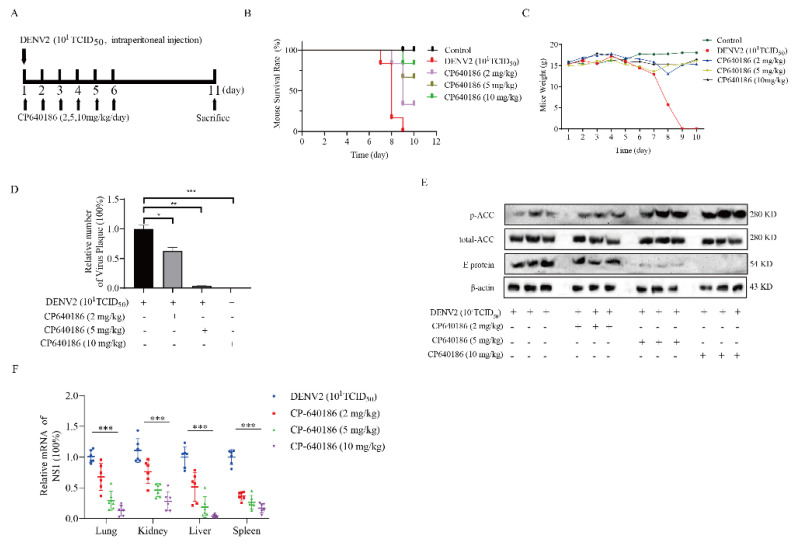
The antiviral effect of CP640186 in vivo. (**A**) Schematic diagram of mice infection assay. IFNAR^−/−^ mice were infected with DNEV2 (10^1^TCID_50_) at day 1. At the same time, the CP640186 (2 mg/kg/day, 5 mg/kg/day, and 10 mg/kg/day) was administered orally for 6 consecutive days. At day 11, these mice were sacrificed for histological analysis. (**B**) The mice survival rate curve. (**C**) The mice weight curve. (**D**) The inoculum of the virus (number) analysis in mouse serum was analyzed by the plaque focus forming assay. (**E**) The p-ACC and viral E protein levels were analyzed by Western blotting in liver tissue. (**F**) The mRNA level of NS1 in the various mice tissues. * indicates *p* < 0.1. ** indicates *p* < 0.01. *** indicates *p* < 0.001.

**Figure 6 molecules-27-08583-f006:**
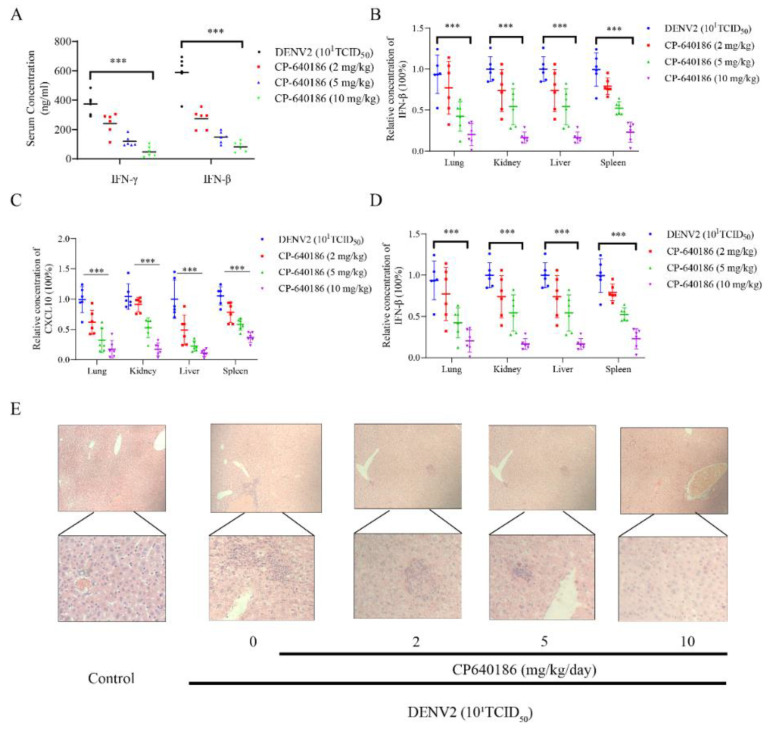
The influence on inflammatory factors of CP640186 in vivo. (**A**) The concentrations of IFN-β and IFN-β were detected by ELISA of the mouse serum in Figure 5A. The mRNA levels of (**B**) IFN-β, (**C**) CXCL10, and (**D**) IL-1β were analyzed by RT-qPCR in Figure 5A. (**E**) The hematoxylin–eosin staining (HE) results of liver tissue: the top figures are 10× magnified, the bottom figures are 40×. *** indicates *p* < 0.001.

## Data Availability

Not applicable.

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
