# Peer review of "Acetyl-CoA Carboxylase (ACC) Inhibitor, CP640186, Effectively Inhibited Dengue Virus (DENV) Infection via Regulating ACC Phosphorylation"

_molecules, 2022, doi:10.3390/molecules27238583_

Round 1
Reviewer 1 Report (Previous Reviewer 1)
I suggest the acceptance in the current form.
Author Response
Thank you for your appreciation
Reviewer 2 Report (New Reviewer)
This manuscript by Wenyu Wu et al. demonstrated that the acetyl-CoA carboxylase (ACC) inhibitor CP640186 reduced dengue virus (DENV) infection in cell cultures and in a mouse model. They also showed that the level of ACC phosphorylation correlated reversely with DENV replication. Three main points are as follows:
1. Accumulated publications have indicated that replication of flaviviruses, including DENV, relies on fatty acid metabolism. A key step within fatty acid metabolism, the carboxylation of acetyl-CoA by ACC to produce malonyl-CoA, has been demonstrated critical in replication of flaviruses (including DENV) in vitro as well as West Nile virus (WNV) in a mouse model (19, 20 in the submitted manuscript).
Therefore, this submitted manuscript does not appear to show adequate originality because a proof of concept with ACC inhibitors as viable antiviral candidates have been reported.
2. The introduction can be improved if current knowledge and significance of the phosphorylation in ACC is addressed. The authors did not further investigate the mechanisms underlying the observed increase in AAC phosphorylation and the attenuated DENV replication (Fig. 2). Nor did they elaborate this issue in the discussion section. This submitted manuscript does not appear to provide new insights into the related topic.
3.There are many typo errors as well as grammatical mistakes that need professional editing.
Author Response
This manuscript by Wenyu Wu et al. demonstrated that the acetyl-CoA carboxylase (ACC) inhibitor CP640186 reduced dengue virus (DENV) infection in cell cultures and in a mouse model. They also showed that the level of ACC phosphorylation correlated reversely with DENV replication. Three main points are as follows:
- Accumulated publications have indicated that replication of flaviviruses, including DENV, relies on fatty acid metabolism. A key step within fatty acid metabolism, the carboxylation of acetyl-CoA by ACC to produce malonyl-CoA, has been demonstrated critical in replication of flaviruses (including DENV) in vitro as well as West Nile virus (WNV) in a mouse model (19, 20 in the submitted manuscript).
Therefore, this submitted manuscript does not appear to show adequate originality because a proof of concept with ACC inhibitors as viable antiviral candidates have been reported.
Response: We partially agree with this comment. This work is based on our previous work about the role of VEGFR2/AMPK in dengue virus replication. Considering ACC is located downstream of AMPK. Based on this result, we proposed the hypothesis that regulating of ACC can inhibit viral infection, so it is theoretically innovative. Although some ACC inhibitors was reported owing the antiviral ability, they are not classical ACC inhibitors. Another problem is that there is only in vivo validation with WNN infected mice model other than dengue virus. Our results applied the classical ACC inhibitor, CP640186, and we systematically evaluated its antiviral activity in mice model infected with dengue virus. So far, no report has studied the ability of ACC inhibitor against dengue virus in vitro and in vivo, and we think this is where our innovation lies.
- The introduction can be improved if current knowledge and significance of the phosphorylation in ACC is addressed. The authors did not further investigate the mechanisms underlying the observed increase in AAC phosphorylation and the attenuated DENV replication (Fig. 2). Nor did they elaborate this issue in the discussion section. This submitted manuscript does not appear to provide new insights into the related topic.
Response: Thanks. The current knowledge of ACC was added in the introduction part.
As for the mechanism studies mentioned, we currently believe that it is mainly due to the inactivation of AMPK during dengue virus infection from our previous study, which thus decreased the phosphorylation of ACC and increased the activity of ACC. While inhibition of ACC could increase the phosphorylation of ACC and reduced the activity of ACC. We added these descriptions to the discussion section. In conclusion, our study provides in vivo data of CP640186 against dengue virus, which is beneficial to the field.
3.There are many typo errors as well as grammatical mistakes that need professional editing.
Response: Thanks. We have revised typo errors and grammatical mistakes all through the revised manuscript.
Reviewer 3 Report (New Reviewer)
This is a nicely written paper about the first report of an ACC inhibitor with in vivo anti-DENV activity.
In the abstract, the authors say that they arrived at CP-640186 ‘after screening’; the authors need to describe the screening process or leave ‘after screening’ out of the abstract and describe how they arrived at CP-640186. Included in this required description, the authors need to say where CP-640186 came from, describe how far it advanced in the drug development process and say why development was halted (including description of important side effects).
Lines 335 through 358 describe how it was already known that ACC inhibitors have antiviral activity and that CP-640186 was known to be an ACC inhibitor. All of this information should be included in the introduction, not in the discussion.
L282-285 – This line needs to be rewritten as it suggests that this report qualifies a compound for ‘clinical usage’. The authors could remove ‘clinical usage’ or ‘until now’, but I suggest rewriting the sentence such that it says that this is the first report of an ACC inhibitor with in vivo anti-DENV activity.
L359 – Not sure what the last sentence means; it should be removed.
Author Response
This is a nicely written paper about the first report of an ACC inhibitor with in vivo anti-DENV activity.
In the abstract, the authors say that they arrived at CP-640186 ‘after screening’; the authors need to describe the screening process or leave ‘after screening’ out of the abstract and describe how they arrived at CP-640186. Included in this required description, the authors need to say where CP-640186 came from, describe how far it advanced in the drug development process and say why development was halted (including description of important side effects).
Lines 335 through 358 describe how it was already known that ACC inhibitors have antiviral activity and that CP-640186 was known to be an ACC inhibitor. All of this information should be included in the introduction, not in the discussion.
Response: Thanks. Based on the reviewer’s kind suggestions, we have rewritten the abstract to make it clearer to understand. Also, we removed the description of CP640186 from the discussion part to the introduction part in the revised manuscript.
L282-285 – This line needs to be rewritten as it suggests that this report qualifies a compound for ‘clinical usage’. The authors could remove ‘clinical usage’ or ‘until now’, but I suggest rewriting the sentence such that it says that this is the first report of an ACC inhibitor with in vivo anti-DENV activity.
Response: Thanks. The sentence was rewritten in the revise manuscript.
L359 – Not sure what the last sentence means; it should be removed.
Response: This sentence was removed from the revised manuscript.
Round 2
Reviewer 2 Report (New Reviewer)
In L342-345, the authors claimed that the previously used AAC inhibitors, PF-05175157, PF-05206574, and PF-06256254, were not good enough. Unfortunately, the authors have not included any of the previously used ACC inhibitors as controls in their experiments to support their arguments. In their responses to the referee, the authors pointed out that the previously used ACC inhibitors are not classical ACC inhibitors. However, they have not addressed why classical ACC inhibitors such as CP640186 used in this study, is superior to those ACC inhibitors that are not classical.
The authors stated that they have reported the important role of AMPK in DENV replication (8 in the submitted manuscript) and that this is a follow-up study of their previous finding. However, this concept regarding AMPK-dengue interactions has been reported way back in 2017 (PMID: 28298606), and is not an original finding from the authors.
Both DENV and WNV viruses are flaviviruses that share common replication strategies, including the dependence on fatty acid metabolism demonstrated in this and previous studies (28 in the submitted manuscript). The ACC inhibitors have been shown to inhibit replication of DENV in vitro as well as that of WNV in vitro and in vivo; therefore, the finding that another ACC inhibitor CP640186 exerted the antiviral activity against DENV in vivo is not of sufficient novelty.
This manuscript is a resubmission of an earlier submission. The following is a list of the peer review reports and author responses from that submission.
Round 1
Reviewer 1 Report
Review
Title: Acetyl-CoA carboxylase (ACC) inhibitor, CP640186, effectively inhibited dengue virus (DENV) infection through regulating lipid metabolism
This work by Wu et al. has investigated and evaluated the effect of CP640186 which is an ACC inhibitor on DENV infection.
Dengue fever is an acute infectious disease caused by dengue virus (DENV) infection. DENV is a vector (mosquito)-borne virus and approximately tens of millions of patients were estimated to exist. DENV could be divided into four serotypes, and there were no approved vaccine nor antivirals on hand.
Based on above background, authors evaluated if CP640186, one of the acetyl-CoA carboxylase (ACC) inhibitor, could exhibit anti-DENV infection.
Overall, the study was well organized and showed that CP640186 possesses anti-DENV effect both in vitro and in vivo, there are several aspects to be clarified and improved. In addition, reviewer requests to submit the manuscript to the English editing service to extensively improve its English throughout the manuscript. Especially, how to use the conjunction must be improved. In addition, there are too many typos or mistakes. Reviewer strongly recommend to check the accuracy of the manuscript before the submission.
Major points:
1. Figure 2D: To evaluate the E protein expression level, quantification such as using western blot, is required. For the quantification, measure fluorescent from the IFA or detecting band intensity from the western blot is common methods.
2. The results showing the plaques are not necessary. Please remove the picture and show only the number after the calculation.
3. Figure 4, to assess the effect of CP640186 on the post-entry step, detection should be stopped at up to 24 h p.i., when the single cycle infection was completed. If the authors detected the virus from 48 h p.i., it has already been passed several virus infection cycle which includes production, re-entry, replication (2nd cycle) etc.. Therefore, similar experiment but the detection period was changed should be conducted.
4. For the references, several articles reported the relationship between the lipid-metabolism and the virus replication. Even only in the positive strand-virus, HCV and lipid droplet/inhibitors against the lipid metabolism has been reported (Miyanari et al., Nat. Cell Biol., 2007, Uchida et al., Viruses, 2016, Raini et al., Antiviral Res., 2021 etc.). Authors should carefully cite and overviewed these articles.
5. In the first part of the discussion, authors described they performed the screening, but it was not described in the text at all. Authors should clarify this point.
6. Throughout the manuscript, authors showed that CP640186 inhibited DENV replication in vitro and in vivo. They also examined the state of ACC phosphorylation. However, they did not assess the effect of CP640186/DENV infection on the lipid metabolism, therefore the title should be reconsidered accordingly.
Minor points:
1. Correct the reference number such as 13 and 14. Current 14 must be 13, and 15 and 16 must be 14 and 15. Go over the remaining as well.
2. Page 2, line 96, 4 strains must be 4 serotypes. Virus strain is not equal to virus strains. Same for page 3, line 98.
3. Page 5, Figure 2, figure legend, (B) and (C) seems to be inversed.
4. Page 6, line 152. “Different dosage viruses” should be better described “different virus dosages”.
5. The expression of “virus titer” should be replaced. Virus titer means virus infectivity. Page 6, line 155, I would recommend from “virus titer increased” to “inoculum of the virus (number) increased”.
6. Page 6, line 160, E protein ⇒ reduction of E protein expression
7. Page 6, line 164, what does “there” mean in this sentence?
8. Page 8, line 222, B6 could mislead the readers. Authors used IFNa/brKO mice which is B6 background. It should be described precisely in the main text not only in the materials and methods. Same description could be adapted in page 10, line 257, please describe as B6 “background” mice.
9. Page 9, Figure 5. Description of when the samples for western blot were collected from the infected mice should be in the text and the materials and methods.
10. Page 9, line 245, Cytokine storm is more common than the inflammatory factor storms.
11. Page 10, line 278. In general, term “transgenic” is used when the gene is introduced in trans. On the other hand, when the gene is deleted, term “knock out (KO)” is used.
12. Terms “in vivo” and “in vitro” should be described in italic (page 10, line 279 etc.).
13. Page 12, line 353, please correct the “Celsius”.
14. Page 12, line 355, For the description of the MTT, please describe in a sentence.
15. Page 12, line 357, rewrite to “BHK-21 cells were seeded … and incubated overnight”. The cells were not seeded overnight, it was incubated overnight.
16. Page 12, line 367, make sure about the 150 ul of DMSO.
17. Page 16, line 400, beta seems to be deleted.
18. Page 16, line 403-404, please describe the detail information and condition for the 2nd antibody reactions.
19. Page 13, line 407, Fix with 1.2%...should be the cells were fixed with 1.2% CMC and incubated for….
20. Page 13, lines 409 and 411, please describe by what Triton and BSA were diluted. PBS -)?
21. Page 13, line 424, please not start with the verb.
22. Page 14, line 441 and 442, IFN-r seems to be doublet.
23. Page 14, line 450, conjunction or subject of the sentence seems to be required. Same for line 453, and page 1 line 14 (in the abstract).
Author Response
Response to the Reviewer’s comments
Thanks to the reviewer’s valuable comments, we have revised the manuscript in accordance with the comments.
Major points:
- Figure 2D: To evaluate the E protein expression level, quantification such as using western blot, is required. For the quantification, measure fluorescent from the IFA or detecting band intensity from the western blot is common methods.
Response: Thanks for your comments. The quantification of Fig. 2D was analyzed by Image J and was shown in the Figure 2F. The bands intensity of the western blot in this study were all quantified through Image J and were shown in the supplemental figure (Figure S1).
- The results showing the plaques are not necessary. Please remove the picture and show only the number after the calculation.
Response: Thanks for your suggestion. All of the plaques figures were removed out from the revised manuscript (Figure S2).
- Figure 4, to assess the effect of CP640186 on the post-entry step, detection should be stopped at up to 24 h p.i., when the single cycle infection was completed. If the authors detected the virus from 48 h p.i., it has already been passed several virus infection cycle which includes production, re-entry, replication (2ndcycle) etc.. Therefore, similar experiment but the detection period was changed should be conducted.
Response: This is a good suggestion. We tried several time points to study the antiviral mode of CP640186 at this dosage of virus, such as 24 hpi, 48 hpi and 72 hpi. In 24 hpi, the E protein level signal was too weak to be detected via the western blotting assay with 10TCID50 of DENV2. While the E protein level signal can be clearly detected in 48 hpi, thus we choose this condition in the study. Despite this, it was still able to observe the inhibition results of the early effects after 48 hpi. We believe that increased viral dosage can observe the corresponding results at 24 hpi. To be consistent with the work of the entire article, we choose this condition.
- For the references, several articles reported the relationship between the lipid-metabolism and the virus replication. Even only in the positive strand-virus, HCV and lipid droplet/inhibitors against the lipid metabolism has been reported (Miyanari et al., Nat. Cell Biol., 2007, Uchida et al., Viruses, 2016, Raini et al., Antiviral Res., 2021 etc.). Authors should carefully cite and overviewed these articles.
Response: Thanks for the reminder. We carefully read these literatures and cited it in the suitable place.
- In the first part of the discussion, authors described they performed the screening, but it was not described in the text at all. Authors should clarify this point.
Response: Thanks, these data was not shown in the manuscript, because we think these results was not critical during preparing this manuscript. To avoid misleading, we have deleted this description in the Discussion section in the revised manuscript.
- Throughout the manuscript, authors showed that CP640186 inhibited DENV replication in vitro and in vivo. They also examined the state of ACC phosphorylation. However, they did not assess the effect of CP640186/DENV infection on the lipid metabolism, therefore the title should be reconsidered accordingly.
Response: Thanks. The lipid droplet was detected in Figure 2B indeed. Even so, maybe we have no more evidence in lipid metabolism, so we respect the reviewer’s comments to narrow the title as “Acetyl-CoA carboxylase (ACC) inhibitor, CP640186, effectively inhibited dengue virus (DENV) infection via regulating ACC phosphorylation” to better match the content.
Minor points:
- Correct the reference number such as 13 and 14. Current 14 must be 13, and 15 and 16 must be 14 and 15. Go over the remaining as well.
Response: Thanks. We have corrected these mistakes in the new version of manuscript.
- Page 2, line 96, 4 strains must be 4 serotypes. Virus strain is not equal to virus strains. Same for page 3, line 98.
Response: Thanks. We have changed the descriptions in the new version of manuscript.
- Page 5, Figure 2, figure legend, (B) and (C) seems to be inversed.
Response: Thanks very much. We have corrected these mistakes in the new version of manuscript.
- Page 6, line 152. “Different dosage viruses” should be better described “different virus dosages”.
Response: Thanks. We have corrected these mistakes in the revised manuscript.
- The expression of “virus titer” should be replaced. Virus titer means virus infectivity. Page 6, line 155, I would recommend from “virus titer increased” to “inoculum of the virus (number) increased”.
Response: We have changed the descriptions in the revised manuscript.
- Page 6, line 160, E protein ⇒reduction of E protein expression
Response: Thanks. We have changed the descriptions in the revised manuscript.
- Page 6, line 164, what does “there” mean in this sentence?
Response: Thanks. We have changed the descriptions in the revised manuscript.
- Page 8, line 222, B6 could mislead the readers. Authors used IFNa/brKO mice which is B6 background. It should be described precisely in the main text not only in the materials and methods. Same description could be adapted in page 10, line 257, please describe as B6 “background” mice.
Response: Thanks. We have revised these descriptions to make it clear in the revised manuscript.
- Page 9, Figure 5. Description of when the samples for western blot were collected from the infected mice should be in the text and the materials and methods.
Response: Thanks. Immediately after the mouse sacrificed, fresh tissue was removed for western blotting or other analyses. We have added these descriptions to make it clear in the revised manuscript.
- Page 9, line 245, Cytokine storm is more common than the inflammatory factor storms.
Response: Thanks. We have changed these descriptions in the revised manuscript.
- Page 10, line 278. In general, term “transgenic” is used when the gene is introduced in trans. On the other hand, when the gene is deleted, term “knock out (KO)” is used.
Response: Thanks.
- Terms “in vivo” and “in vitro” should be described in italic (page 10, line 279 etc.).
Response: Thanks.
- Page 12, line 353, please correct the “Celsius”.
Response: Thanks.
- Page 12, line 355, For the description of the MTT, please describe in a sentence.
Response: Thanks. The MTT assay was described in one sentence.
- Page 12, line 357, rewrite to “BHK-21 cells were seeded … and incubated overnight”. The cells were not seeded overnight, it was incubated overnight.
Response: Thanks.
- Page 12, line 367, make sure about the 150 ul of DMSO.
Response: Thanks. A hole of 96 well plate can hold up to 200 mL of liquid, 150 mL is the actual volume in this study.
- Page 16, line 400, beta seems to be deleted.
Response: Thanks.
- Page 16, line 403-404, please describe the detail information and condition for the 2ndantibody reactions.
Response: Thanks for your suggestion. We have added the detail information in the new version of manuscript.
- Page 13, line 407, Fix with 1.2%...should be the cells were fixed with 1.2% CMC and incubated for….
Response: Thanks.
- Page 13, lines 409 and 411, please describe by what Triton and BSA were diluted. PBS -)?
Response: Thanks. It’s PBS.
- Page 13, line 424, please not start with the verb.
Response: Thanks for your suggestion.
- Page 14, line 441 and 442, IFN-r seems to be doublet.
Response: Thanks. We have corrected this mistake.
- Page 14, line 450, conjunction or subject of the sentence seems to be required. Same for line 453, and page 1 line 14 (in the abstract).
Response: Thank you for your comments. Since we are non-native speakers, we have already invited native speakers to revise the whole of this manuscript.
Reviewer 2 Report
The manuscript is poorly written and the experiments are not well designed. There are many inconsistencies between the text and the figures (eg lines 225 infected mice die by day 8, and fig 1B shows death by day 9; line 228 indicates that 10 mg treatment all mice survive and the figure does not show that; lines 236 talk about a variety of organs analized and the figure presents just liver data, etc). The number of animals used in the in vivo experiments is not known. The legends of the figures do not correspond in many occasions with what is shown
Author Response
Response to the Reviewer’s comments
Thanks to the reviewer’s valuable comments, we have revised the manuscript in accordance with the comments.
The manuscript is poorly written and the experiments are not well designed. There are many inconsistencies between the text and the figures (eg lines 225 infected mice die by day 8, and fig 1B shows death by day 9; line 228 indicates that 10 mg treatment all mice survive and the figure does not show that; lines 236 talk about a variety of organs analyzed and the figure presents just liver data, etc). The number of animals used in the in vivo experiments is not known. The legends of the figures do not correspond in many occasions with what is shown.
Response: Thanks for your valuable comments. Based on your comments, we have tried our best to revise the manuscript thoroughly, we hope that our revision can make the reader know what we are doing.
Fistly, for the comments about the timing of death of mice, all infected mice that did not received CP640186 treatment were sacrisfied from day7 to day8, which all scarified at the 8th day, and the other survival mice were all sacrificed on the 11th day. This description was added into the revised manuscript.
Secondly, we recalculated the survival rate of mice and revised these description in the manuscript. In the 2 mg oral treatment group, the survival rate of mice was 34%. In the 5 mg oral treatment group, the survival rate of mice was 66%. In the 10 mg treatment group, all nearly 84% mice survived (Figure 5B). We have revised these descriptions in the new version of manuscript.
Actually, a variety of organs analyzed by RT-qPCR, such as lung, kidney and spleen. If it refers to the western results, we have not put it in the manuscript for the consideration of the figure length. These results were added in the supplemental part of new version of manuscript (Figure S3).
As for the other questions you raised, we have thoroughly checked and revised them in the revised version.
Round 2
Reviewer 1 Report
Revised manuscript was improved well, but still need some corrections for the English such as Zika agent (line 67, page 2) to Zika virus.
Reviewer 2 Report
The manuscript doesn´t reach enough quality to be considered for publication in this journal.
Author Response
Response to the Reviewer’s comments
The manuscript doesn´t reach enough quality to be considered for publication in this journal.
Response: We have improved the quality of this manuscript thoroughly and we hope it’s suitable for this journal.